# Myelodysplastic Syndromes/Myeloproliferative Overlap Neoplasms and Differential Diagnosis in the WHO and ICC 2022 Era: A Focused Review

**DOI:** 10.3390/cancers15123175

**Published:** 2023-06-13

**Authors:** Diletta Fontana, Elena M. Elli, Fabio Pagni, Rocco Piazza

**Affiliations:** 1Department of Medicine and Surgery, University of Milano-Bicocca, 20900 Monza, Italy; diletta.fontana@unimib.it; 2Hematology Division and Bone Marrow Unit, Fondazione IRCCS San Gerardo dei Tintori, 20900 Monza, Italy; elenamaria.elli@irccs-sangerardo.it; 3Department of Medicine and Surgery, Pathology, Fondazione IRCCS San Gerardo dei Tintori, 20900 Monza, Italy; fabio.pagni@unimib.it

**Keywords:** WHO 2022, ICC, MDS/MPN, CMML, myelodysplastic/myeloproliferative neoplasm with neutrophilia, aCML, myelodysplastic/myeloproliferative neoplasm, *SF3B1* mutation, thrombocytosis, ring sideroblasts

## Abstract

**Simple Summary:**

Myelodysplastic syndromes/myeloproliferative neoplasms (MDS/MPN) are entities that have been quite difficult to define since their discovery. At the time of presentation, they possess features of both myelodysplastic syndromes and myeloproliferative neoplasms. Because of this overlap, there has been an inherent difficulty in the diagnosis and classification of these neoplasms. The recent World Health Organization (WHO) 2022 classification and the International Consensus Classification (ICC) improved the diagnostic criteria for these disorders. In this review, we describe the main entities, highlighting the differential diagnosis.

**Abstract:**

The myelodysplastic syndromes/myeloproliferative neoplasms (MDS/MPN) category comprises a varied group of myeloid neoplastic diseases characterized by clinical and pathologic overlapping features of both myelodysplastic and myeloproliferative neoplasms. For these reasons, these tumors are challenging in terms of diagnosis. The recent World Health Organization (WHO) 2022 classification and the International Consensus Classification (ICC) made changes in the classification of MDS/MPN compared to the previous 2016 WHO classification and improved the diagnostic criteria of these entities. The aim of this review is to describe the main entities reported in the more recent classifications, focusing on chronic myelomonocytic leukemia (CMML), MDS/MPN with neutrophilia (or atypical CML [aCML]), and MDS/MPN with *SF3B1* mutation and thrombocytosis/MDS/MPN with ring sideroblasts and thrombocytosis. A particular emphasis is given to the differential diagnosis and analysis of subtle divergences and semantic differences between the WHO classification and the ICC for these entities.

## 1. Introduction

The myelodysplastic syndromes/myeloproliferative overlap neoplasms (MDS/MPN) comprise a heterogeneous group of myeloid neoplastic diseases with clinical and pathologic overlapping features of both myelodysplastic and myeloproliferative neoplasms [1]. The 2016 World Health Organization (WHO) classification included five entities: chronic myelomonocytic leukemia (CMML), atypical CML *BCR-ABL1^−^* (aCML), juvenile myelomonocytic leukemia (JMML), MDS/MPN with ring sideroblasts and thrombocytosis (MDS/MPN-RS-T), and MDS/MPN unclassifiable (MDS/MPN-U) [2] (Table 1). In 2022, there emerged two competing classifications for myeloid neoplasms: the International Consensus Classification (ICC) and the fifth edition of the WHO classification [3,4]. Both classifications now expand on these categories; in particular, MDS/MPN with ring sideroblasts and thrombocytosis (MDS/MPN-RS-T) has been split into two entities in the ICC 2022 based on the presence/absence of the *SF3B1* mutation. Moreover, both the WHO 2022 classification and the ICC move JMML to be grouped with pediatric and/or germline mutation-associated disorders (Table 1).

In MDS/MPN neoplasms the myeloproliferative component coexists together with the ineffective hematopoiesis, leading to cytopenia. In the WHO 2022 classification [3], the two most important keywords used to describe MDS/MPN diseases are “cytopenia”, together with “cytosis”. Only a small percentage of early-stage CMML patients are borderline or have no cytopenia. In these patients, bone marrow morphology, flow cytometric profiling, and molecular analyses are required to support the diagnosis [4].

Notably, a monocytosis (>10%, and >0.5 × 10^9^/L of the WBC) may identify conditions with the potential to progress to MDS/MPN in patients with clonal cytopenia of undetermined significance (CCUS). Thus, based on persistent monocytosis together with myeloid mutations, and in the absence of BM morphologic findings of CMML, the ICC identifies the condition of clonal monocytosis of undetermined significance (CMUS) which, in the presence of cytopenia, becomes a “clonal cytopenia with monocytosis” [4].

In this review, we discuss the recently updated morphologic and molecular diagnostic criteria of MDS/MPN, focusing more on CMML, myelodysplastic/myeloproliferative neoplasm with neutrophilia (named yet aCML in the ICC), and MDS/MPN with *SF3B1* mutation and thrombocytosis/MDS/MPN with ring sideroblasts and thrombocytosis.

## 2. Chronic Myelomonocytic Leukemia

CMML is the most common (incidence 0.6 × 100,000 people in the United States) MDS/MPN disease. It is a clonal stem cell disorder that is characterized by sustained peripheral blood (PB) monocytosis (≥0.5 × 10^9^/L and ≥10% of white blood cells [WBC] differential count) and an inherent tendency for transformation to acute myeloid leukemia (AML; 15% over 3 to 5 years) [5].

The median age at diagnosis for CMML is 73 to 75 years, with a male preponderance. CMML diagnosis remains largely based on morphology. Histologically, CMML was classified as CMML-0 (<2% PB blasts and <5% bone marrow [BM] blasts), CMML-1 (2–4% PB blasts and/or 5–9% BM blasts), and CMML-2 (5–19% PB blasts and/or 10–19% BM blasts or when Auer rods are present) [6]. However, recent studies have shown that the prognostic impact of CMML-0 and CMML-1 is virtually identical. In addition, the classification of CMML patients as CMML-0 or CMML-1 requires accurate counting of a low number of blast cells. This is particularly challenging in the case of CMML, in which blasts include promonocytes, whose distinction from abnormal monocytes can at times be problematic. For these reasons, the WHO 2022 classification reverted to the fourth edition 2-tiered system of CMML-1 (<5% blasts in PB, <10% in BM) and CMML-2 (5–19% blasts PB, 10–19% in BM, or Auer rods), therefore incorporating CMML-0 cases into the CMML-1 group [4].

Based on the presenting WBC count, the French–American–British (FAB) classification distinguished CMML into two subtypes: myeloproliferative CMML (MP-CMML; WBC count ≥ 13 × 10^9^/L) and myelodysplastic CMML (MD-CMML; WBC count < 13 × 10^9^/L), where the former has a poor outcome and a higher rate of AML transformation [7]. In CMML, the proliferative component is manifested as monocytosis often in association with splenomegaly and/or leukocytosis [8]. Several prognostic models exist for CMML; however, the CMML-specific prognostic scoring system–molecular model (CPSS molecular score) is the principal one [7]. This includes the percentage of BM blasts, the FAB subtype of CMML (WBC > 13 × 10^9^/L), the need for transfusion support, and the presence of genetic markers (*ASXL1, RUNX1, NRAS, SETBP1,* and cytogenetic abnormalities) as risk factors.

### 2.1. Cytogenetics and Molecular Genetics

CMML is a clonal disorder predominantly arising in the context of age-related clonal hematopoiesis [9]. Clonal cytogenetic abnormalities occur in ~30% of patients, with a variability largely due to small numbers, inclusion criteria, and referral patterns. CMML is not characterized by specific cytogenetic aberrations; however, trisomy 8 and monosomy 7 are the most frequent, while complex karyotypes are infrequent [10]. A strong association between specific cytogenetic abnormalities (trisomy 8, chromosome 7 abnormalities, or complex karyotype) and the risk of AML evolution and overall survival (OS) is described [11].

In the coding regions of CMML patients’ genome, an average of 10 to 15 somatic mutations can be found [12]. Compared to MDS and AML the mutational spectrum of CMML is more homogeneous [8]. The most common somatic mutations involve splicing genes and epigenetic modifiers (e.g., *SRSF2, TET2*, and/or *ASXL1*), which occur in about 80% of cases [5]. Other mutated genes include *SETBP1, NRAS/KRAS, RUNX1, CBL,* and *EZH2*. The prototypical molecular fingerprint combines a mutation in a gene encoding an epigenetic regulator (mainly *TET2* and *ASXL1*) with a mutation affecting the spliceosome machinery (*SRSF2*, less often *SF3B1*, *ZRSR2*) with or without a mutation in the RAS/MAPK signaling pathway [8]. Overall, >90% of patients with CMML would be expected to show at least one of these mutations with modern next generation sequencing (NGS) techniques [13,14]. Mutations involving *TET2* (epigenetic) and *SRSF2* (splicing) often skew hematopoiesis toward monocytosis, with subsequent mutations in epigenetic regulators (*ASXL1*) or signaling pathways (*NRAS*, *CBL*, *KRAS*, *PTPN11*, and *JAK2*), giving rise to a full-blown disease [6].

MPN-CMML variant is enriched in active RAS/MAPK signaling, with ~70% of patients demonstrating RAS pathway mutations (e.g., *NRAS*, *KRAS*, *CBL*), with *NRAS* being the most commonly involved RAS-family gene. RAS pathway mutations, along with epigenetic events, also play a role in CMML transformation to AML. Of note, *NPM1* mutation is seen in a rare subset of CMML (3–5%) [15]. The development of an *NPM1* mutation in the setting of a known CMML should be noted, as it bears prognostic relevance. It is important to note that, although the new WHO classification states that the presence of the *NPM1* variant alone leads to a diagnosis of AML independently from the blast percentage [4], such a finding in the ICC does not automatically define de novo AML in the setting of known CMML, as a bone marrow blast count of at least 10% is also required [4]. Regardless of these differences in the WHO classification and the ICC, CMML *NPM1* mutated appears to herald a particularly aggressive clinical course with a median overall survival of 12.5 months compared to 20.5 months for *NPM1* wild-type patients [15]. Therefore, this subgroup of CMML *NPM1* mutated probably deserves an AML-like therapeutic approach.

A recommended minimal NGS panel for CMML [8] is represented by the following genes: *TET2* (frequency: 29–61%), *ASXL1* (frequency: 32–44%), *DNMT3A* (frequency: 2–12%), *EZH2* (frequency: 5–13%), *IDH1* (frequency: 1–2%), *IDH2* (frequency: 6–7%), *BCOR* (frequency: 6–7%), *SRSF2* (frequency: 29–52%), *U2AF1* (frequency: 4–10%), *SF3B1* (frequency: 6–10%), *ZRSR2* (frequency: 4–8%), *CBL* (frequency: 8–22%), *KRAS* (frequency: 7–16%), *NRAS* (frequency: 4–22%), *NF1* (frequency: 6–7%), *JAK2* (frequency: 1–10%), *RUNX1* (frequency: 8–23%), *SETBP1* (frequency: 4–18%), *NPM1* (frequency: 1–3%), and *FLT3* (frequency: 1–3%) [7,8,9,12,16,17,18,19,20,21,22,23,24,25,26,27,28].

### 2.2. Flow Cytometry Immunophenotyping

CMML diagnosis can benefit from flow cytometry analyses of BM and PB cells. Indeed, the detection of subtle changes in the surface antigen expression of myelomonocytic cells and the erythroid lineage can potentially facilitate the monitoring of the disease [8]. In particular, flow cytometry analysis of monocyte subsets readily distinguishes CMML from benign reactive monocytosis in patients with PB monocytosis ≥ 1 × 10^9^/L. As reported by the current nomenclature of normal human monocyte subsets, the fraction of classical monocytes (CD14^+^/CD16^−^), also known as MO1, can be distinguished from intermediate, MO2 monocytes expressing CD14^+^ and CD16^+^, and from nonclassical, MO3 monocytes, typically expressing CD16 and low levels of CD14 [29]. This is important in the context of CMML, as the proportion of MO1 is increased in CMML patients, while it is decreased in those affected by a reactive disorder [30]. In particular, a MO1 ≥ 94% provides a very high level of specificity and sensitivity to distinguish CMML from reactive monocytosis [29,30].

### 2.3. Diagnosis and Differential Diagnosis of CMML

The revised diagnostic criteria according to the WHO 2022 classification are presented in Table 2 and include prerequisite and supporting criteria. The first prerequisite criterion is persistent absolute (≥0.5 × 10^9^/L) and relative (≥10%) peripheral blood monocytosis, in the absence of reactive etiologies. Namely, the cutoff for absolute monocytosis is lowered from 1.0 × 10^9^/L to 0.5 × 10^9^/L, to incorporate cases formerly referred to as oligomonocytic CMML. The second prerequisite is to have a PB and BM blast count of less than 20%, with or without BM dysplasia. The third and fourth prerequisites require not meeting the diagnostic criteria of chronic myeloid leukemia or other myeloproliferative neoplasms, and of myeloid/lymphoid neoplasms with tyrosine kinase fusions, respectively. In line with these exclusion criteria, CMML diagnosis also requires the absence of molecular aberrations, such as *PDGFRA, PDGFRB,* and *PCM1::JAK2*, that can be associated with clonal disorders potentially characterized by the presence of monocytosis.

Supporting criteria are instead represented (1) by the presence of dysplasia in one or more than one lineage (Figure 1A), (2) by the evidence of clonality, typically demonstrated by the identification of somatic mutations or cytogenetic abnormalities, and (3) by the evidence of abnormal partitioning of peripheral blood monocyte subsets (Figure 1B). The latter is introduced as a new supporting criterion, with most CMML patients demonstrating an expansion (>94%) of classical monocytes (MO1^−^CD14^+^/CD16^−^). This modality has also shown promise in distinguishing CMML from other causes of monocytosis, including MPN with monocytosis, and it is useful in cases of CMML without evidence of clonality [29,30]. In the presence of clear evidence of absolute monocytosis, i.e., with monocytes ≥ 1.0 × 10^9^/L, all prerequisite criteria must be present and further strengthened by at least one of the supporting criteria. To enhance diagnostic accuracy when absolute monocytosis is low, i.e., between 0.5 × 10^9^/L and 1.0 × 10^9^/L, the detection of one or more clonal cytogenetic or molecular abnormalities and the documentation of dysplasia in at least one lineage are required.

In the new WHO classification, the presence of somatic mutations as a supporting criterion is felt to be at least as critical as the presence of dysplasia for confirming a diagnosis of CMML. Indeed, having established the need for clonality as one of the supporting criteria, the evidence of dysplasia becomes necessary only for those rare patients who lack the presence of CMML-associated mutations or when monocytosis is low, as already discussed. The rationale for this approach is based on three considerations: (1) the frequent evidence of clonal variants in CMML, which makes the complete lack of mutations highly unlikely; (2) the availability of myeloid NGS panels able to reliably detect subclonal mutations down to the 1–5% variant allele frequency (VAF) range; (3) the challenges associated with the objective demonstration of dysplasia in bone marrow specimens. In addition, mutations in CMML have prognostic implications; hence, the implementation of mutation detection in the diagnostic routine of CMML allows for the killing of two birds with one stone, on the one hand, and confirming the diagnosis on the other, to refine the prognosis.

An absolute monocytosis is uncommon in classical CML; however, it may be present in a subset of patients with MPN and can be detected at the time of primary diagnosis or during the disease progression. Monocytosis is also associated with a more unfavorable outcome in patients affected by polycythemia vera (PV) [31]. Monocytosis is also a powerful and independent predictor of inferior survival in primary myelofibrosis (PMF) [32]. Among classical driver mutations, in 10% of CMML patients, JAK2V617F can occur, while mutations in MPL and CALR genes are extremely infrequent and their detection should raise questions with regard to a bona fide CMML diagnosis. In addition, the multiparametric flow cytometry (discussed below) can help differentiate CMML from MPN with monocytosis [33].

Triple-negative primary myelofibrosis (TN-PMF) and myelodysplastic syndromes with fibrosis (F-MDS) are rare entities, which are often difficult to distinguish and occasionally associated with a CMML-like phenotype. Currently, no specific molecular markers are available, and the integration of clinical data, such as BM morphology and blood counts, remains essential for diagnosis. Of note, neutrophil granulopoiesis, age-related cellularity, and changes in erythropoiesis, together with the severity of BM fibrosis should be considered in addition to megakaryocytic atypia (Figure 1C). NGS tests might be useful to distinguish between both entities and to refine prognosis. TN-PMF and F-MDS show a high rate of mutations in myeloid genes, with TET2, U2AF1, SETBP1, TP53, or RUNX1 being more frequently mutated in F-MDS. TN MF presents poor outcomes and a high risk of leukemic transformation [34].

Mutations involving SRSF2, SETBP1, IDH2, CBL, and GNAS are significantly enriched in TN disease.

Reactive monocytosis is very common in clinical practice and the principal causes are represented by viral infections, recovery from injury, drugs or chemotherapy, bacterial subacute endocarditis, tuberculosis, brucellosis, leishmaniosis, and autoimmune diseases such as systemic lupus erythematosus, sarcoidosis, and mixed connective tissue disorder. Reactive monocytosis can also be seen in the context of metastatic solid neoplasms [8].

## 3. Clonal Monocytosis of Undetermined Significance and Clonal Cytopenia with Monocytosis of Undetermined Significance

The ICC recognizes the clonal monocytosis of undetermined significance (CMUS), a CMML precursor condition, as an independent entity [4]. The CMUS was proposed in order to classify a particular disorder that does not yet fulfill the criteria of CMML [35]. Indeed, the CMUS is characterized by persistent monocytosis (monocytes ≥ 10% and≥ 0.5 × 10^9^/L of the WBC), the presence of myeloid neoplasm-associated mutation(s), and the absence of BM morphologic findings of CMML (Table 3). In cases of cytopenia, the disorder is renamed clonal cytopenia and monocytosis of undetermined significance (CCMUS) [4].

## 4. MDS/MPN with Neutrophilia (Also Known as aCML)

The fifth edition of the WHO Classification has updated the name of atypical chronic myeloid leukemia, *BCR::ABL1*-negative (aCML), and replaced it with myelodysplastic syndrome/myeloproliferative neoplasms with neutrophilia (MDS/MPN with neutrophilia) [3], while the ICC has kept the original name [4]. This change highlights the MDS/MPN nature of the disease, while the absence of the terms “atypical CML” and “*BCR::ABL1*-negative” in the WHO classification avoids potential confusion with both classical CML and other Philadelphia-negative disorders [3]. Diagnostic criteria according to the WHO 2022 classification and the ICC for this entity are presented in Table 4.

MDS/MPN with neutrophilia is a rare clonal hematopoietic stem cell disorder of the elderly, with a median age at presentation between 60 and 76 years [36]. Its estimated incidence is 1 out of 100 cases of t(9;22)(q34;q11), *BCR::ABL1*-positive CML, meaning approximately 1 case per 1,000,000 persons per year [37,38,39,40]. Although a female predominance or no sex predominance has been reported in early studies [38,41], in more recent years, a slight male predominance has been evidenced in reports that analyze larger cohorts of patients [42,43,44]. MDS/MPN with neutrophilia is a disorder that presents clinical features similar to *BCR::ABL1*-positive CML, including splenomegaly and neutrophilic leukocytosis. Moreover, this leukemia predominantly affects the neutrophilic lineage associated with neutrophilic leukocytosis and circulating immature granulocytic precursors constituting ≥10% of all leukocytes [45]. MDS/MPN with neutrophilia also shows prominent granulocytic dysplasia (e.g., hypogranular and hypolobated neutrophils, abnormal chromatin clumping, and pseudo Pelger–Huet neutrophils) [46]. For these reasons, a main differential diagnosis of MDS/MPN with neutrophilia is *BCR::ABL1*-positive CML. To detect the presence of *BCR::ABL1* translocation, the most therapeutically relevant diagnostic test is karyotype analysis, completed with molecular testing, such as reverse transcription polymerase chain reaction (RT-PCR) and fluorescence in situ hybridization (FISH) techniques, which are both able to reveal the presence of cryptic Ph translocations [47]. The lack of the *BCR::ABL1* fusion gene prevents treatment with tyrosine kinase inhibitors, such as imatinib or second- or third-generation Abelson inhibitors. Therefore, its prognosis remains very poor, with a median overall survival of 24 months from diagnosis [48].

Since the clinical features of MDS/MPN with neutrophilia overlap with other myeloproliferative and myelodysplastic malignancies, the diagnosis is still challenging and relies primarily on morphologic criteria. Indeed, besides the absence of the Philadelphia chromosome and of the *BCR::ABL1* translocation, the presence of dysgranulopoiesis in BM or PB is the major criterion that can be used to distinguish it from *BCR::ABL1*-positive CML.

In MDS/MPN with neutrophilia, basophilia is not as prominent as in CML, as basophils represent <2% of all leukocytes [2,46,49]; the diagnostic hallmark for aCML, as defined by the ICC and WHO classification, is the presence of leukocytosis ≥ 13 × 10^9^/L, with ≥10% of immature granulocytes and <20% blasts in the PB and the BM [2,5,46,50]. Of note, ICC requires PB cytopenias, with similar thresholds as MDS, for a diagnosis of aCML [4].

In MDS/MPN with neutrophilia, it is also common to observe an absolute monocyte count > 1 × 10^9^/L; however, the percentage of monocytes at onset must be lower than 10% of the total leukocytes. This characteristic is crucial to discriminate MDS/MPN with neutrophilia from CMML, as in CMML, the presence of both absolute and relative monocytosis is required for diagnosis [1,2]. The proportion of immature myeloid cells (promyelocytes, myelocytes, and metamyelocytes) in PB, as well as the presence of dysplasia, are key criteria for the differential diagnosis against a rare *BCR::ABL1*-negative myeloproliferative neoplasm named chronic neutrophilic leukemia (CNL), as in CNL, the dysplasia is absent, the percentage of immature elements is always <10%, and persistent peripheral blood neutrophilia (WBC > 25 × 10^9^/L, with >80% segmented neutrophils plus banded neutrophils) is present [2,40]. The higher frequency of *CSF3R* mutations in CNL [51] has been proposed as an additional criterion to orient the diagnosis; however, *CSF3R* mutations can be seen also in MDS/MPN with neutrophilia with highly variable frequency in different studies [52,53,54,55]. Therefore the real impact of *CSF3R* variants for the differential diagnosis of the two disorders is currently a matter of debate [56].

MDS/MPN with neutrophilia exhibits a hypercellular BM with myeloid hyperplasia and prominent granulocytic dysplasia (Figure 2A,B). In addition, trilineage dysplasia may be present [1,52,57]. Conversely, it is not characterized by eosinophilia; hence, eosinophils are expected to be less than 10% in the differential count [4,58]. This feature usually makes the separation between MDS/MPN with neutrophilia and chronic eosinophilic leukemia, not otherwise specified (CEL, NOS), quite straightforward [59]. In addition, as myeloid/lymphoid neoplasms with eosinophilia and tyrosine kinase gene fusions (MLN-TK) are characterized by genetic abnormalities such as *PDGFRA*, *PDGFRB*, *FGFR1*, *JAK2*, *FLT3* rearrangements, or tyrosine kinase fusions (e.g., *ETV6::ABL1*, *ETV6::FGFR2*, *ETV6::LYN*, *ETV6::NTRK3*, *RANBP2::ALK*, *BCR::RET*, *FGFR1OP::RET*) [3], their differential diagnosis is largely based on molecular features.

Regarding the myeloproliferative disorders, it is known that *JAK2*, *CALR*, and *MPL* genes, which are usually associated with MPN disorders such as PMF, PV, and essential thrombocythemia (ET), have to be absent in MDS/MPN with neutrophilia [60]. On the one hand, the absence of mutations occurring in these genes can support the differentiation between MDS/MPN with neutrophilia and PMF, but this is usually the most arduous (Figure 2C); on the other hand, the identification of the rare myeloproliferative cases negative for all the three marker genes, named triple-negative myeloproliferative disorders, becomes very difficult. In such cases, genetic analysis can facilitate the diagnosis. Indeed, as reported by the 2016 WHO revision [2], in up to one-third of cases, MDS/MPN with neutrophilia is characterized by the presence of recurrent mutations in *ETNK1* (3.7–13.3%) [61,62,63] and *SETBP1* (7.4–48%) [64,65,66] genes, often in association with *ASXL1* (20–81%), whose presence has been linked to a more aggressive disease [4,36,43,58,67,68]. Besides these genes, other frequent somatic mutations involve *NRAS* and *KRAS* (11–27%), *SRSF2* (14–65%), *EZH2* (19–30%), *RUNX1* (11–15%), *TET2* (27–33%), and *CBL* (8–11%) [36,43,48,50,52,58,61,64,66,68,69,70,71,72,73,74,75,76,77]. According to NGS analysis, the fifth edition of the WHO criteria outlines supportive somatic mutations, such as *ASXL1* and *ETNK1* for diagnosis [3]. In contrast, the ICC has also added molecular supportive criteria, focusing on the presence of somatic mutations involving *ASXL1* and *SETBP1* [4].

## 5. MDS/MPN with *SF3B1* Mutation and Thrombocytosis and MDS/MPN with Ring Sideroblasts and Thrombocytosis

The “myelodysplastic syndrome/myeloproliferative neoplasm with ring sideroblasts and thrombocytosis (MDS/MPN-RS-T)” in the 2016 WHO classification [2] is renamed MDS/MPN with *SF3B1* mutation and thrombocytosis (MDS/MPN-T) in the WHO 2022 classification [3], and split into two entities in the ICC [4], represented by MDS/MPN with thrombocytosis and *SF3B1* mutation (MDS/MPN-T *SF3B1*), and MDS/MPN with ring sideroblasts and thrombocytosis, not otherwise specified, in the absence of *SF3B1* mutation (MDS/MPN RS-T, NOS), as reported in Table 1.

Key elements of this clonal disorder were in the WHO 2016 classification for the presence of dysplasia (Figure 3A–D), either involving the sole erythroid lineage and causing anemia, or involving multiple lineages, with >15% ring sideroblasts and coexisting with thrombocytosis. From a molecular point of view, the most relevant element is the frequent co-occurrence of somatic *SF3B1* mutations, found in approximately 80% of the MDS/MPN-T cases, together with the JAK2 V617F mutation, found in 50–60%, or less commonly, with *CALR* or *MPL* mutations, globally found in less than 10% of the MDS/MPN-T cases. Clonal cytogenetic abnormalities are less common, as they are found in 20% of cases.

As mutations occurring in the spliceosome gene *SF3B1* are known to be enriched in patients affected by MDS/MPN-RS-T, the identification of ring sideroblasts is not considered to be required to confirm the diagnosis in the most recent WHO classification [3], as well as in the ICC [4], provided that *SF3B1* somatic mutation is found with a VAF greater than 10%.

According to the WHO 2022 classification [3], the term MDS/MPN with ring sideroblasts and thrombocytosis has been kept as an acceptable term to be used for cases with wild-type *SF3B1* and ≥15% ring sideroblasts. Conversely, this entity is separated from *SF3B1* mutated MDS/MPN forms in the ICC and designed as “MDS/MPN with sideroblasts and thrombocytosis, NOS”.

In agreement with the criteria defined for other MDS/MPN, and for MDS/MPN-T-*SF3B1* and MDS/MPN-RS-T NOS, both thrombocytosis and anemia must be present at onset [4].

The ICC 2022 diagnostic criteria for the two variants MDS/MPN-T *SF3B1* and MDS/MPN RS-T, NOS are presented in Table 5 and Table 6, respectively.

Differential diagnosis for MDS/MPN-T comprises myeloproliferative disorders, MDS, and other MDS/MPN neoplasms. Differential diagnosis against ET, sharing with MDS/MPN-T the presence of thrombocytosis, can be based on morphological as well as molecular criteria. From a morphological point of view, in the presence of mutated *SF3B1*, the evidence of anemia, optionally associated with erythroid dysplasia in the bone marrow, definitely supports the diagnosis of MDS/MPN-T. In the absence of *SF3B1*, anemia must be associated with erythroid-lineage dysplasia and ring sideroblasts to support the diagnosis of MDS/MPN-RS-T, NOS. At the molecular level, the evidence of *SF3B1* mutations, either in the presence or absence of activating *JAK2* mutations, determines the diagnosis of MDS/MPN-T *SF3B1*, regardless of the identification of ring sideroblasts, if *SF3B1* is detectable with VAF > 10%. In contrast, the presence of *JAK2*, *CALR,* or *MPL* mutations in the absence of *SF3B1* should orient the diagnosis to a myeloproliferative disorder.

Finally, cases of MDS-*SF3B1* that later develop thrombocytosis are now considered to represent the thrombocytotic progression of MDS-*SF3B1*.

## 6. Myelodysplastic/Myeloproliferative Neoplasm, Not Otherwise Specified

MDS/MPN, unclassifiable is now termed MDS/MPN, not otherwise specified MDS/MPN, NOS) in both the WHO classification and the ICC [3,4]. This fact is in line with the international effort to remove the qualifier “unclassifiable” from the classification. MDS/MPN, NOS is still mainly a diagnosis of exclusion; however, it is now better described by the inclusion of diagnostic requirements that are reported in Table 7. These criteria include a need for the presence of cytopenia in association with myeloproliferative features in PB and a lack of specific gene rearrangements/fusions of M/LN-Eo with tyrosine kinase gene fusions. Therefore, although the diagnosis can be made in the absence of clonality or mutations, even in the case of histopathologic evidence of MDS/MPN and the exclusion of other MDS/MPN entities, the establishment of clonality is expected [4].

## 7. Discussion

The family of MDS/MPN disorders comprises a heterogeneous group of neoplasms characterized by the co-existence of myelodysplastic and myeloproliferative features. The recent WHO 2022 classification [3] and the International Consensus Classification (ICC) [4] now expand on these categories and move JMML to the group of pediatric and/or germline mutation-associated disorders (Table 1). Both classifications exclude JMML from the category of overlap neoplasms, since JMML is a pediatric-onset entity with germline predisposition, typically characterized by the presence of germline *PTPN11*, *CBL*, and *NF1* mutations and by predominantly myeloproliferative features. While essential features for the diagnosis of overlap neoplasms have been maintained by both classification schemata, subtle differences, with the potential to impact patient care, are present [5].

Numerous mutations in signaling genes, such as *CBL, JAK2, NRAS, KRAS, CSF3R*, and others involving the spliceosome complex have been identified in MDS/MPN disorders, therefore supporting the notion of MDS/MPN as heterogeneous clonal disorders. These observations suggest that the co-mutation of genes involved in dysplasia and bone marrow failure along with those of cytokine receptor signaling may, in part, explain the dual MDS/MPN phenotype. The respective MDS/MPN diseases are identified by the type of myeloid subset that predominates in the peripheral blood. The aim of this review was to describe the main entities reported in the more recent classifications, focusing on the principal overlap of MDS/MPN neoplasms.

CMML is the most frequent and is characterized by the presence of sustained peripheral blood monocytosis with recurrent mutations involving *TET2*, *SRSF2,* and *ASXL1*; with RAS pathway mutations and *JAK2*V617F being relatively enriched in proliferative CMML subtypes (WBC ≥ 13 × 10^9^/L). CMML usually occurs around the age of 70, with a male preponderance, and a median overall survival of <36 months. The presence of mutations in splicing genes and epigenetic modifiers as a means to demonstrate clonality is felt to be critical for confirming a diagnosis of CMML, but the WHO classification and the ICC also underline the role of flow cytometry analysis in cases without clonality: here, the atypical monocyte subset distribution readily distinguishes CMML from benign reactive monocytosis.

MDS/MPN with neutrophilia results in dysplastic neutrophilia in the absence of monocytosis and eosinophilia and is associated with a high rate of transformation to AML. *ASXL1* mutations are frequent, with the disease displaying a higher prevalence of *ETNK1*, *SETBP1*, and *EZH2* mutations in comparison to other MDS/MPN overlap neoplasms.

Finally, MDS/MPN-T, (both MDS/MPN-T-*SF3B1* and MDS/MPN-RS-T, NOS) are MDS/MPN overlap syndromes characterized by features of MDS, typically including RS, dyserythropoiesis, and thrombocytosis, as seen in MPN. From a clinical point of view, anemia and thrombocytosis are common, with BM showing atypical megakaryocytes and dysplastic erythrocytes, usually with rare blasts. Mutational profiles are similar to those of ET and MDS: the most common mutation seen in up to 90% of MDS/MPN-T patients occurs in *SF3B1,* which is now considered to be a driver mutation of this disorder. Other most common mutations occur in the *JAK2* gene, as well as in other genetic/epigenetic regulators, such as *ASXL1*, *TET2*, *DNMT3A*, and *SETBP1.*

## 8. Conclusions

The MDS/MPN myeloid neoplasms comprise a group of disorders with a heterogeneous mutation landscape and overlapping features of both myelodysplastic and myeloproliferative neoplasms.

These characteristics render the diagnosis and correct classification of MDS/MPN neoplasms particularly challenging. In this review, we provide an updated view of these disorders which takes into account the contribution of the new WHO and ICC classifications. We hope that this reasoned approach, which focused on the differential diagnosis of these elusive clonal pathologies, will help clinicians in their diagnostic workup.

## Figures and Tables

**Figure 1 cancers-15-03175-f001:**
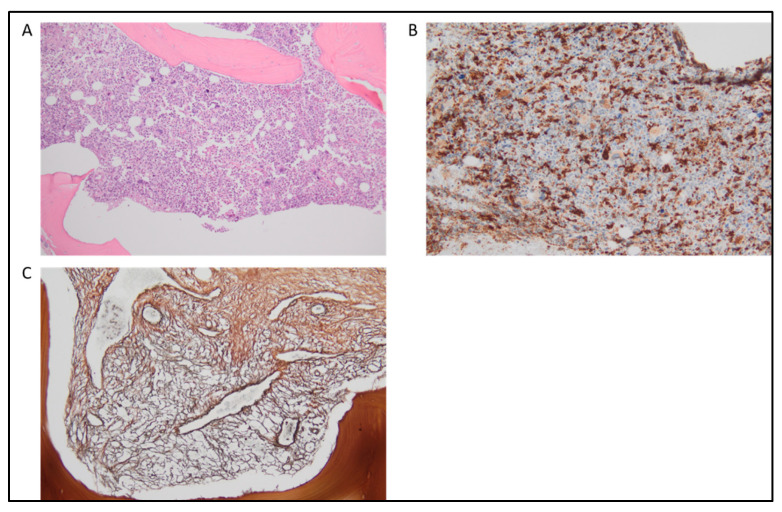
Chronic myelomonocytic leukemia. (**A**) H&E staining (20×) multilineage dysplasia with irregular megakaryocytes characterized by hypolobulated and hyperchromatic nuclei. (**B**) A predominant monocyte differentiation is highlighted by CD68 pgm1 immunohistochemical staining (20×). (**C**) At silver staining (20×) large bands of extensive fibrosis (MF2, WHO score 2018).

**Figure 2 cancers-15-03175-f002:**
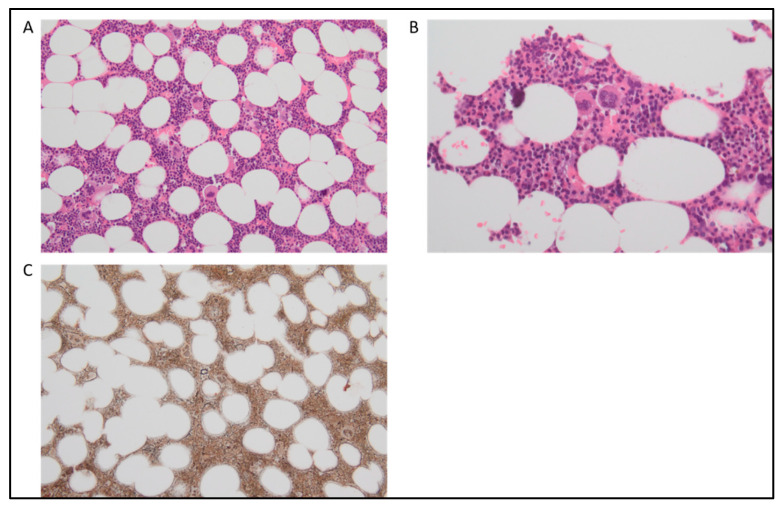
MDS/MPN with neutrophilia. (**A**) At H&E (20×), a mildly hypercellular bone marrow shows minimal megakaryocytes dysplasia. No significant prevalence of immature cells in the other lines. (**B**) H&E at 40×. (**C**) Silver staining demonstrates a mild increase of the reticular fibers (MF1, WHO score 2018).

**Figure 3 cancers-15-03175-f003:**
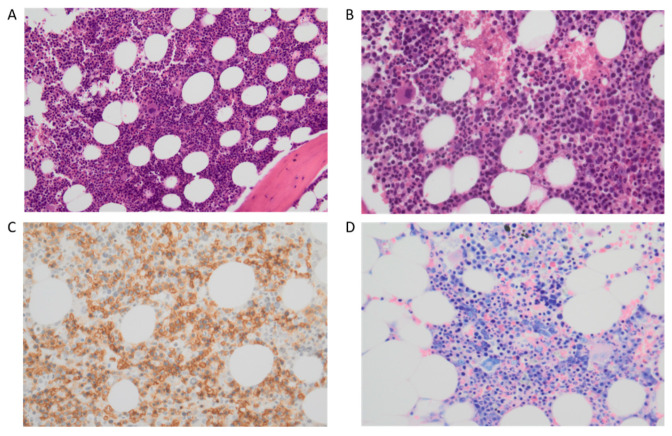
MDS/MPN with *SF3B1* mutation and thrombocytosis. (**A**–**C**) At H&E ((**A**): 20×; (**B**): 40×), moderate multilineage dysplasia with significant dyserythropoiesis and megalbostastosis, as demonstrated by the glycophorin staining (**C**). (**D**) At MGG, other dysplastic findings in the erythroid series, which correspond to Perls’ positive ring sideroblasts at bone marrow aspirate.

**Table 1 cancers-15-03175-t001:** Myelodysplastic/myeloproliferative neoplasms according to the WHO 2016 [2], WHO 2022 [3], and ICC 2022 [4] classifications.

WHO 2016 Classification	WHO 2022 Classification	ICC 2022 Classification
Chronic myelomonocytic leukemia	Chronic myelomonocytic leukemia	Chronic myelomonocytic leukemia
		Clonal cytopenia with monocytosis of undetermined significance Clonal monocytosis of undetermined significance
Atypical chronic myeloid leukemia (aCML), *BCR-ABL1^−^*	Myelodysplastic/myeloproliferative neoplasm with neutrophilia	Atypical chronic myeloid leukemia
Juvenile myelomonocytic leukemia (JMML)		
MDS/MPN with ring sideroblasts and thrombocytosis (MDS/MPN-RS-T)	Myelodysplastic/myeloproliferative neoplasm with *SF3B1* mutation and thrombocytosis	Myelodysplastic/myeloproliferative neoplasm with thrombocytosis and *SF3B1* mutation
		Myelodysplastic/myeloproliferative neoplasm with ring sideroblasts and thrombocytosis, not otherwise specified
MDS/MPN, unclassifiable	Myelodysplastic/myeloproliferative neoplasm, not otherwise specified	Myelodysplastic/myeloproliferative neoplasm, not otherwise specified

Adapted from [2,3,4].

**Table 2 cancers-15-03175-t002:** Diagnostic criteria for chronic myelomonocytic leukemia (CMML) according to the WHO 2022 classification [3].

Prerequisite Criteria
1. Persistent absolute (≥0.5 × 10^9^/L) and relative (≥10%) peripheral blood monocytosis
2. Blasts constitute < 20% of the cells in the peripheral blood and bone marrow ^a^
3. Not meeting diagnostic criteria of chronic myeloid leukemia or other myeloproliferative neoplasms ^b^
4. Not meeting diagnostic criteria of myeloid/lymphoid neoplasms with tyrosine kinase fusions ^c^
**Supporting criteria**
1. Dysplasia involving ≥1 myeloid lineages ^d^
2. Acquired clonal cytogenetic or molecular abnormality
3. Abnormal partitioning of peripheral blood monocyte subsets ^e^
**Requirements for diagnosis**
- Pre-requisite criteria must be present in all cases
- If monocytosis is ≥1 × 10^9^/L: one or more supporting criteria must be met
- If monocytosis is ≥0.5 and <1 × 10^9^/L: supporting criteria 1 and 2 must be met
**Subtyping criteria**
- Myelodysplastic CMML (MD-CMML): WBC < 13 × 10^9^/L
- Myeloproliferative CMML (MP-CMML): WBC ≥ 13 × 10^9^/L
**Subgrouping criteria** (based on percentage of blasts and promonocytes)
CMML-1: <5% in peripheral blood and <10% in bone marrow
CMML-2: 5–19% in peripheral blood and 10–19% in bone marrow

^a^ Blasts and blast equivalents include myeloblasts, monoblasts, and promonocytes. ^b^ Myeloproliferative neoplasms (MPN) can be associated with monocytosis at presentation or during the course of the disease; such cases can mimic CMML. In these instances, a documented history of MPN excludes CMML. The presence of MPN features in the bone marrow and/or high burden of MPN-associated mutations (*JAK2*, *CALR,* or *MPL*) tends to support MPN with monocytosis rather than CMML. ^c^ Criteria for myeloid/lymphoid neoplasms with tyrosine kinase fusions should be specifically excluded in cases with eosinophilia. ^d^ Morphologic dysplasia should be present in ≥10% of cells of a hematopoietic lineage in the bone marrow. ^e^ Based on the detection of increased classical monocytes (>94%) in the absence of known active autoimmune diseases and/or systemic inflammatory syndromes. Adapted from [3].

**Table 3 cancers-15-03175-t003:** Diagnostic criteria for clonal monocytosis of undetermined significance according to the ICC [4].

Diagnostic Criteria
Persistent monocytosis defined as monocytes > 0.5 × 10^9^/L and >10% of the WBC
Absence or presence of cytopenia (thresholds same as for MDS) ^a^
Presence of at least one myeloid neoplasm-associated mutation of appropriate allele frequency (i.e., ≥2%) ^b^
No significant dysplasia, increased blasts (including promonocytes), or morphologic findings of CMML on bone marrow examination ^c^
No criteria for a myeloid or other hematopoietic neoplasm are fulfilled
No reactive condition that would explain a monocytosis is detected

^a^ If cytopenia is present the nomenclature of clonal cytopenia and monocytosis of undetermined significance (CCMUS) is suggested. ^b^ VAF threshold based on International Consensus Group Conference, Vienna, 2018 [35]. ^c^ Bone marrow findings of CMML include hypercellularity with myeloid predominance, often with increased monocytes and in a proportion of cases, monoblasts and/or blast equivalents (i.e., promonocytes) and/or dysplasia in at least one lineage. Adapted from [4].

**Table 4 cancers-15-03175-t004:** Diagnostic criteria for MDS/MPN with neutrophilia according to the WHO 2022 classification [3] and the ICC [4,5].

Criteria	ICC 2022 Classification	WHO 2022 Classification
Nomenclature	Atypical chronic myeloid leukemia	Myelodysplastic/myeloproliferative neoplasm with neutrophilia
White blood cell count	≥13 × 10^9^/L with immature ^a^ myeloid cells constituting ≥ 10% of WBC	≥13 × 10^9^/L with neutrophilia, with immature ^a^ myeloid cells constituting ≥10% of WBC
Cytopenia	MDS ^b^ -qualifying thresholds	Not specifically mentioned in the WHO criteria
Peripheral blood and bone marrow blasts	<20%	<20%
Dysplasia	Dysgranulopoiesis; hyposegmented or hypersegmented neutrophils, with or without abnormal chromatin clumping	Circulating immature ^a^ myeloid cells constituting ≥ 10% of WBC, with neutrophilic dysplasia
Eosinophils	<10%	Not specifically mentioned
Monocytes	<10%	<10%
Bone marrow cellularity and hematopoiesis	Hypercellular with granulocytic hyperplasia and granulocytic dysplasia, with or without involvement of other lineages	Hypercellular with granulocytic hyperplasia and granulocytic dysplasia, with or without involvement of other lineages
Molecular exclusionary criteria	*BCR::ABL1* or tyrosine kinase fusions associated with myeloid/lymphoid neoplasms with eosinophilia. *JAK2*, *MPL*, and *CALR* mutations	*BCR::ABL1* or tyrosine kinase fusions associated with myeloid/lymphoid neoplasms with eosinophilia. *JAK2*, *MPL*, and *CALR* mutations. *CSF3R* mutations MDS/MPN-RS-T with *SF3B1* mutations
Next generation sequencing data ^c^	Desirable to document the presence of *ASXL1* and SETBP1 mutations.	Desirable to document the presence of *SETBP1* and/or ETNK1 mutations

^a^ Immature myeloid cells include promyelocytes, myelocytes, and metamyelocytes. ^b^ MDS-defining cytopenias include Hb < 13 g/dL in males and <12 g/dL in females, neutropenia with absolute neutrophil count < 1.8 × 10^9^/L, and thrombocytopenia with platelet counts < 150 × 10^9^/L. ^c^ Supportive and desirable criteria. Adapted from [3,4,5].

**Table 5 cancers-15-03175-t005:** Diagnostic criteria for MDS/MPN-T *SF3B1*, according to the ICC [4].

Diagnostic Criteria
Thrombocytosis, with platelet count ≥ 450 × 10^9^/L
Anemia (threshold same as for MDS)
Blasts < 1% in blood and <5% in bone marrow
Presence of *SF3B1* mutation (VAF > 10%), isolated or associated with abnormal cytogenetics and/or other myeloid neoplasm-associated mutations
No history of recent cytotoxic or growth factor therapy that could explain the myelodysplastic/myeloproliferative features
No *BCR::ABL1* or genetic abnormalities of myeloid/lymphoid neoplasms with eosinophilia and tyrosine kinase gene fusions; no t(3;3)(q21.3;q26.2), inv(3)(q21.3q26.2), or del(5q) *
No history of MPN, MDS, or other myelodysplastic/myeloproliferative neoplasm

* In a case that otherwise meets the diagnostic criteria for MDS with del(5q). Adapted from [4].

**Table 6 cancers-15-03175-t006:** Diagnostic criteria for MDS/MPN-RS-T, NOS, according to the ICC [4].

Diagnostic Criteria
Thrombocytosis, with platelet count ≥ 450 × 10^9^/L
Anemia associated with erythroid-lineage dysplasia, with or without multilineage dysplasia, and ≥15% ring sideroblasts
Blasts < 1% in blood and <5% in bone marrow
Presence of clonality: demonstration of a clonal cytogenetic abnormality and/or somatic mutation(s). In their absence, no history of recent cytotoxic or growth factor therapy that could explain the myelodysplastic/myeloproliferative features.
Absence of *SF3B1* mutation; no *BCR::ABL1* or genetic abnormalities of myeloid/lymphoid neoplasms with eosinophilia and tyrosine kinase gene fusions; no t(3;3)(q21.3;q26.2), inv(3) (q21.3q26.2), or del(5q) *
No history of MPN, MDS, or other MDS/MPN

* In a case that otherwise meets the diagnostic criteria for MDS with del(5q). Adapted from [4].

**Table 7 cancers-15-03175-t007:** Diagnostic criteria for myelodysplastic/myeloproliferative neoplasm, NOS, according to the ICC [4].

Diagnostic Criteria
Myeloid neoplasm with mixed myeloproliferative and myelodysplastic features, not meeting the WHO criteria for any other myelodysplastic/myeloproliferative neoplasm, myelodysplastic syndrome, myeloproliferative neoplasm ^a^
Cytopenia (thresholds same as for MDS)
Blasts <20% of the cells in blood and bone marrow
A platelet count of ≥450 × 10^9^/L and/or a white blood cell count of ≥13 × 10^9^/L
Presence of clonality: demonstration of a clonal cytogenetic abnormality and/or somatic mutation(s). If clonality cannot be determined, the findings have persisted and all other causes (e.g., history of cytotoxic or growth factor therapy or other primary cause that could explain the myelodysplastic/myeloproliferative features) have been excluded.
No *BCR::ABL1* or genetic abnormalities of myeloid/lymphoid neoplasms with eosinophilia and tyrosine kinase gene fusions; no t(3;3)(q21.3;q26.2), inv(3)(q21.3q26.2), ^b^ or del(5q) ^c^

^a^ Myeloproliferative neoplasms (MPNs), in particular those in the accelerated phase and/or the post-polycythemia vera or post-essential thrombocythemia myelofibrotic stages, may simulate MDS/MPN, NOS. A history of MPN and/or the presence of MPN-associated mutations (in *JAK2*, *CALR,* or *MPL*), particularly if associated with a high VAF, tend to exclude a diagnosis of MDS/MPN, NOS. The presence of hypereosinophilia would favor a diagnosis of CEL, NOS. ^b^ In a case that otherwise meets the criteria for MDS-NOS. ^c^ In a case that otherwise meets the diagnostic criteria for myelodysplastic syndrome with isolated del(5q). Adapted from [4].

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
