# Peer review of "Myelodysplastic Syndromes/Myeloproliferative Overlap Neoplasms and Differential Diagnosis in the WHO and ICC 2022 Era: A Focused Review"

_cancers, 2023, doi:10.3390/cancers15123175_

Round 1

Reviewer 1 Report

In the present review article, Fontana et al describe the main entities reported in WHO-2022 classifications, such as CMML, aCML, and MDS/MPN with SF3B1 mutation and thrombocytosis/MDS/MPN with ring sideroblasts and thrombocytosis and compared it with ICC-2020 classification and WHO-2016 classification. The review article is well-written and most of the criteria are considered and well defined in paper However, some issues should be addressed:

1)     When comparing recent WHO and ICC-2022 with WHO 2016 classification, it would be better to add WHO 2016 classification as well in Table 1 and describe in brief.

2)     To strengthen the paper, it would be better if they compared the differences between all of the diseases (such as Clonal cytopenia with monocytosis of undetermined

Significance and Myelodysplastic/myeloproliferative neo-plasm not otherwise specified) mentioned in Table 1 in the ICC-2022 classification instead of just describing the four diseases used by WHO-2022.

3)     There is a description of flow markers for CMML disease. To improve the strength of the papaer, it would be better to discuss it in the discussion as well as flow markers are used for diagnosis of the disease along with the molecular mutation.

Author Response

We are glad that the three Reviewers found merits in our work and we did our best to address their points. The point-by-point answers to the Reviewers are reported here below.

Answers to Reviewer #1

In the present review article, Fontana et al describe the main entities reported in WHO-2022 classifications, such as CMML, aCML, and MDS/MPN with SF3B1 mutation and thrombocytosis/MDS/MPN with ring sideroblasts and thrombocytosis and compared it with ICC-2020 classification and WHO-2016 classification. The review article is well-written and most of the criteria are considered and well defined in paper However, some issues should be addressed:

1)     When comparing recent WHO and ICC-2022 with WHO 2016 classification, it would be better to add WHO 2016 classification as well in Table 1 and describe in brief.

We thank the Reviewer for this suggestion. To better clarify our work, WHO 2016 classification was added in Table 1, and a brief description was added in the Introduction.

2)     To strengthen the paper, it would be better if they compared the differences between all of the diseases (such as Clonal cytopenia with monocytosis of undetermined Significance and Myelodysplastic/myeloproliferative neo-plasm not otherwise specified) mentioned in Table 1 in the ICC-2022 classification instead of just describing the four diseases used by WHO-2022.

In agreement with the Reviewer, sections entitled “Clonal Monocytosis of Undetermined Significance and Clonal Cytopenia with Monocytosis of Undetermined Significance” and “Myelodysplastic/myeloproliferative neoplasm, not otherwise specified” were added to Table 1 and a brief description of them was added in the Introduction and in the text.

3)     There is a description of flow markers for CMML disease. To improve the strength of the paper, it would be better to discuss it in the discussion as well as flow markers are used for diagnosis of the disease along with the molecular mutation.

In agreement with the reviewer, a sentence about the role of flow cytometry was added in discussion.

Reviewer 2 Report

This is a well-organized review of the MDS/MPN overlap disorders with focus on classification vs treatment. 

Comments:

1  Lines 14 and 15; The second sentence is confusing.  ? state "They possess features of both MDS and MPN syndromes."

2. Line 91--would specifiy what the FAB classification of CMML is and in fact what FAB is and why this matters 

3, Line 289--the disease CNL is suddenly introduced.  Would define this for the reader as well as where it fits in the WHO/ICC classifications. 

4 . LInes 402 and 403 are important.  How can pathologists present reports to clinicians that encompass these classifications with different nomenclature to accurately define disease states to inform correct treatment and prognosis discussions?   

Minor:

Line 60 and 272: morphological should be morphologic 

LIne 95--contest should be context

LInes 128 and 129--no new paragraph needed. 

Author Response

We are glad that the three Reviewers found merits in our work and we did our best to address their points. The point-by-point answers to the Reviewers are reported here below.

Answers to Reviewer #2

This is a well-organized review of the MDS/MPN overlap disorders with focus on classification vs treatment. 

Comments:

1  Lines 14 and 15; The second sentence is confusing.  ? state "They possess features of both MDS and MPN syndromes."

We agree. The text was modified as suggested.

  1. Line 91--would specify what the FAB classification of CMML is and in fact what FAB is and why this matters 

We agree. We added a sentence about FAB classification of myelodysplastic and myeloproliferative CMML subtypes in the text.

3, Line 289--the disease CNL is suddenly introduced.  Would define this for the reader as well as where it fits in the WHO/ICC classifications. 

We thank the reviewer for raising this issue, which is critical given the challenges associated with the differential diagnosis of CNL vs MDS/MPN. To overcome this limitation a description of CNL was added to the text, as suggested.

4 . Lines 402 and 403 are important.  How can pathologists present reports to clinicians that encompass these classifications with different nomenclature to accurately define disease states to inform correct treatment and prognosis discussions?   

We thank the reviewer for this important comment. A multidisciplinary approach for different MDS/MPN could help formulate an integrated clinical-histopathological diagnosis, that is more easily accessible to the clinicians in clinical practice; this type of approach would be desirable, but is beyond the scope of this review.

Minor:

Line 60 and 272: morphological should be morphologic 

We thank the Reviewer for pointing this out. The text was modified as suggested.

Line 95--contest should be context

We thank the Reviewer for pointing this out. The text was modified as suggested.

Lines 128 and 129--no new paragraph needed. 

Corrected

Reviewer 3 Report

A very nice, well-written and definitely useful review.

The description is complete and accurate. I have only a minor suggestion. I believe that adding a few images of cytology and histopathology of the described antities might be nice and useful to the readers

Author Response

We are glad that the three Reviewers found merits in our work and we did our best to address their points. The point-by-point answers to the Reviewers are reported here below.

Answers to Reviewer #3

A very nice, well-written and definitely useful review.

The description is complete and accurate. I have only a minor suggestion. I believe that adding a few images of cytology and histopathology of the described entities might be nice and useful to the readers

Thanks for raising this important point. As dysplasia is so critical in this context, adding a few images of the different disorders could indeed improve the overall quality of this review. Therefore, we added images of the principal entities discussed in this review, as suggested.